# Radiomic Features from Post-Operative ^18^F-FDG PET/CT and CT Imaging Associated with Locally Recurrent Rectal Cancer: Preliminary Findings

**DOI:** 10.3390/jcm12052058

**Published:** 2023-03-06

**Authors:** Dajana Cuicchi, Margherita Mottola, Paolo Castellucci, Alessandro Bevilacqua, Arrigo Cattabriga, Maria Adriana Cocozza, Stefano Cardelli, Gerti Dajti, Susanna Mattoni, Rita Golfieri, Stefano Fanti, Alberta Cappelli, Francesca Coppola, Gilberto Poggioli

**Affiliations:** 1Medical and Surgical Department of Digestive, Hepatic and Endocrine-Metabolic Diseases, IRCCS Azienda Ospedaliero-Universitaria di Bologna, 40138 Bologna, Italy; 2Department of Radiology, IRCCS Azienda Ospedaliero-Universitaria di Bologna, 40138 Bologna, Italy; 3Department of Medical and Surgical Sciences (DIMEC), University of Bologna, 40138 Bologna, Italy; 4Department of Nuclear Medicine, IRCCS Azienda Ospedaliero-University of Bologna, 40138 Bologna, Italy; 5Department of Computer Science and Engineering (DISI), University of Bologna, 40126 Bologna, Italy; 6Advanced Research Center on Electronic Systems (ARCES), University of Bologna, 40125 Bologna, Italy; 7Interventional Radiology Unit, IRCCS Azienda Ospedaliero-Universitaria di Bologna, 40138 Bologna, Italy; 8Radiology Unit, “Infermi” Hospital, 48018 Faenza, Italy; 9SIRM Foundation, Italian Society of Medical and Interventional Radiology, 20122 Milano, Italy

**Keywords:** radiomics, locally recurrent rectal cancer, radiomic features, positron emission tomography, computed tomography, image processing, cancer imaging

## Abstract

Locally Recurrent Rectal Cancer (LRRC) remains a major clinical concern; it rapidly invades pelvic organs and nerve roots, causing severe symptoms. Curative-intent salvage therapy offers the only potential for cure but it has a higher chance of success when LRRC is diagnosed at an early stage. Imaging diagnosis of LRRC is very challenging due to fibrosis and inflammatory pelvic tissue, which can mislead even the most expert reader. This study exploited a radiomic analysis to enrich, through quantitative features, the characterization of tissue properties, thus favoring an accurate detection of LRRC by Computed Tomography (CT) and ${}^{18}$F-FDG-Positron Emission Tomography/CT (PET/CT). Of 563 eligible patients undergoing radical resection (R0) of primary RC, 57 patients with suspected LRRC were included, 33 of which were histologically confirmed. After manually segmenting suspected LRRC in CT and PET/CT, 144 Radiomic Features (RFs) were generated, and RFs were investigated for univariate significant discriminations (Wilcoxon rank-sum test, *p* < 0.050) of LRRC from NO LRRC. Five RFs in PET/CT (*p* < 0.017) and two in CT (*p* < 0.022) enabled, individually, a clear distinction of the groups, and one RF was shared by PET/CT and CT. As well as confirming the potential role of radiomics to advance LRRC diagnosis, the aforementioned shared RF describes LRRC as tissues having high local inhomogeneity due to the evolving tissue’s properties.

## 1. Introduction

Over the past three decades, multimodality therapy has significantly improved oncological outcomes of locally advanced rectal cancer. The widespread use and optimisation of Total Mesorectal Excision (TME) and the constant use of neoadjuvant Chemoradiotherapy (nCRT) have sharply decreased the rate of local recurrence after surgery from 20–30% to 5–10% [1]. However, Locally Recurrent Rectal Cancer (LRRC) remains a major clinical concern. Untreated LRRC tends to progress with local invasion of the pelvic organs and nerve roots, causing severe symptoms that affects quality of life, such as intense and refractory pelvic pain, bleeding, rectal or vaginal malodorous discharge, tenesmus, bowel obstruction and or fistulation [2]; LRRC untreated or treated with palliative treatments is usually fatal within 3–12 months [3]. Curative-intent salvage therapy offers the only potential for cure and for preservation of quality of life. The five-year cancer-specific survival after radical resection may reach 45–50% in high volume institutions [4,5]. Salvage surgery clearly has a higher chance of success when LRRC is diagnosed at an early stage [6].

Anastomotic recurrence is easy to identify at clinical evaluation and/or at surveillance proctoscopy or flexible sigmoidoscopy, nevertheless the identification of pelvic recurrence is much more complex. Diagnostic imaging has to differentiate fibrosis and inflammatory tissue from tumour tissue within a pelvis whose anatomy has been altered by previous surgery and radiotherapy. Cross-sectional chest and abdomino-pelvic Ccomputed Tomography (CT) are the surveillance imaging recommended by guidelines to rule out the presence of LRRC [7,8,9]. A pelvic lesion that enlarges on consecutive post-operative CT studies is highly suspicious for LRRC; however, an early diagnosis is not always easy to make. Fluorine-18 2-Fluoro-2-Deoxy-D-Glucose (${}^{18}$F-FDG) Positron Emission Tomography/CT (PET/CT) scan and pelvic Magnetic Resonance Imaging (MRI), although not typically recommended, might be considered for imaging to follow-up abnormalities seen on CT scans [8]. Nowadays, MRI mainly serves as a road map for the surgical procedure and increases the chances of margin clearance; nevertheless, its role in the detection of LRRC is not well established [10]. In addition, FDG avidity in the presacral space at PET/CT not uncommonly proves to be due to benign inflammatory changes. Thus, presacral lesions should be interpreted with caution, and treatment decisions should be made with histopathological confirmation [10]. Sometimes, more biopsies repeated over time are needed to confirm the presence of a recurrence, delaying its diagnosis.

The application of well-established machine learning and artificial intelligence techniques to medical image analysis, nowadays known as radiomics, notably enriches the information retrievable from different types of clinical images (e.g., CT, MR, and PET images) and ultimately improves the diagnostic potential of the imaging modalities. In fact, the Radiomic Features (RFs) extracted from routinely acquired medical images enable a quantitative and objective characterization of tissue properties, latent ones included [11]. Accordingly, RFs become potential promotors of predictive imaging biomarkers, thereby allowing the early detection of LRRC.

To the best of our knowledge, only one report on MRI-based radiomics exists for the assessment of LRRC at the site of anastomosis, but none on pelvic recurrence [12]. Then, the aim of this study is to investigate the potential role of radiomics from post-operative CT and PET/CT images in predicting pelvic recurrence of rectal cancer, identifying from both imaging modalities the RFs which could distinguish patients with and without LRRC.

## 2. Materials and Methods

### 2.1. Study Population

This retrospective study was approved by the local ethics committee (Comitato Etico Area Vasta Emilia Centro, AVEC c/o “IRCCS Azienda Ospedaliero-Universitaria di Bologna” n${}^{{}^{\circ}}$ 848/2020/OSS/AOUBo) and informed consent was waived because of its retrospective nature.

All patients undergoing radical resection (R0, >1 mm resection margin) of the primary rectal adenocarcinoma with or without neoadjuvant therapy from January 2007 to May 2021 at the Division of Colorectal Surgery, “IRCCS Azienda Ospedaliero-Universitaria di Bologna” were considered for inclusion into this study. Inclusion criteria were (1) histologically confirmed LRRC; (2) radiologically suspected but not histologically confirmed LRRC; (3) at least one follow-up examination (abdominal CT or PET/CT) that raised the suspicion of LRRC and that was available at local radiological Picture Archiving and Communication System (PACS). Exclusion criteria were (1) palliative resection of primary tumour; (2) R1 resection of primary tumour (microscopic resection of ≤1 mm, involving margins); (3) R2 resection of primary tumor (resection with macroscopically involved margins); (4) local excision of primary tumor; (5) mucinous adenocarcinoma; (6) patients without histologically confirmed LRRC who did not complete at least 18 months of follow-up; (7) imaging not available at local radiological PACS or studies with corrupted Dicom header.

All patients were evaluated for management by the Multidisciplinary Rectal Cancer Team at “IRCCS Azienda Ospedaliero-Universitaria di Bologna” comprising oncologists, surgeons, radiation oncologists, radiologists, gastroenterologists, a genetic counselor, and pathologists. Data of all patients were prospectively recorded in a dedicated database. Decisions regarding the treatment approach (neoadjuvant therapy versus upfront surgery) of primary tumor were made with reference to the clinical stage and the location of the tumor. The definition of rectum has evolved over time. Until 2013, we considered rectal tumors as those localized within 12 cm from the anal verge at rigid proctoscopy. After that, the rectum was defined by anatomical criteria demonstrated on MRI as being the portion of the large bowel below the sacral promontory that is surrounded by a definable mesorectum posteriorly [10]. Recently we used the “Sigmoid Take-Off” method (STO) as a radiological landmark to identify the anatomical point of transition between the mesorectum and sigmoid mesocolon, and we defined rectal cancer as any tumor with a lower border starting below the STO [13]. According to the principles of TME, standard rectal resection was performed in all cases. Follow-up was performed at six-month intervals during the five post-operative years consisting of physical exams, digital rectal examination, serum tumor-markers Carcinoembryonic Antigen (CEA), endorectal ultrasound when possible, and annual chest and abdomino-pelvic CT scan. Colonoscopy was performed at 6 months (if not complete before surgery), 2 years and 5 years after surgery and every 3–5 years thereafter; patients were referred for pelvic MRI and/or PET/CT when during follow-up there was a clinical suspicion for LRRC. All cases of suspected LRRC were discussed in the Multidisciplinary Rectal Cancer Team meetings and CT images reviewed by two radiologists. Histologic confirmation of recurrent disease was usually required, and tissue biopsy was obtained percutaneously with CT-guidance.

LRRC was defined as recurrence of rectal cancer within the pelvis after previous surgical resection [10]. LRRC includes anastomotic and tumor bed recurrence within lymphatics such as residual mesorectal nodes and pelvic side-wall lymph nodes [10]. Cases where biopsy was positive constitute the LRRC group. Cases without signs of LRRC on consecutive imaging examinations (CT, MRI, and FDG-PET/CT) during a follow-up period of at least 18 months with or without negative biopsy constitute the NO LRRC group. For both groups, patient demographics, tumor characteristics, surgical intervention, and administration of radiotherapy and systemic chemotherapy were reviewed. All patients were followed from the resection of primary tumor to the date of their last follow-up or death.

#### 2.1.1. LRRC Group

According to the Leeds group system, the LRRC location was classified into the following subsites: central (tumor confined to pelvic organs or connective tissue without contact onto, or invasion into, bone), sacral (tumor present in the presacral space and abuts onto or invades the sacrum), sidewall (tumor involving lateral pelvic sidewall structures including greater sciatic foramen and sciatic nerve through to piriformis and the gluteal region), and composite (sacral and sidewall combined) [14]. The extent of the disease was evaluated by chest, abdominal, and pelvis CT, pelvic MRI, and PET/CT scans. The presence of distant metastatic disease and a predicted R2 resection margin were generally considered contraindications to curative–intent salvage therapy. Patients who had not already received pelvic irradiation for their primary tumor underwent preoperative therapy with 50·4 Gy in 28 fractions and concurrent 5-Fluorouracil (5-FU)/capecitabine; patients who had received pelvic irradiation previously were treated with hyperfractionated radiation therapy or surgery. Surgical resection was undertaken with the aim to obtain an R0 resection, and multispecialty surgical teams were assembled in cases of multivisceral resections and for soft tissue reconstruction.

#### 2.1.2. NO LRRC Group

Patients with suspected LRRC without histological confirmation were followed up every 3–6 months for an early detection of tumor relapse. Patients were considered LRRC-free if the size of suspected pelvic lesion did not enlarge in 18 months of follow up without therapy and if biopsies, sometimes even repeated over time, were negative.

The diagram in Figure 1 shows the flowchart of participants in the study.

### 2.2. Computed Tomography Image Protocol Acquisition

CT studies were performed using a 64-section multidetector CT scanner (Lightspeed VCT 64; GE Healthcare, Milwaukee, WI, USA). The scans were carried out using the following parameters: 5 mm section thickness (reconstructed with a 2 mm thickness, with an overlap of 1 mm), pitch of 5.5, 120 kV, and 130–181 mA. The exams were performed with a first unenhanced acquisition followed by the administration of 90–140 mL (according to patient weight) of the tri-iodinated nonionic contrast agent Iomeron® (Bracco, Milan, Italy [350 mg iodine per mL]) at a flow rate of 2–3 mL/s into an antecubital vein by using an automated power injector. The study was acquired in portal-venous phase using bolus tracking; a Region Of Interest (ROI) was placed over the descending aorta and a threshold of 150 HU was selected. Once this had been reached, a multiphasic study was performed with scans acquired in late arterial phase (after 25–30 s), portal venous phase (after 45–60 s), and delayed phase (after 180–300 s). Moreover, multiplanar reformation on sagittal, para-coronal, and para-axial planes, oriented on the rectal axis, might be obtained, provided useful information in most challenging cases, where surgery determined a substantial alteration of normal anatomy.

### 2.3. Positron Emission Tomography Image Protocol Acquisition

Patients underwent clinical routine ${}^{18}$F-FDG PET/CT. After a fasting time of at least 6 h, a single injection in bolus of ${}^{18}$F-FDG (mean adjusted dose 3.5–4.5 MBq/kg) was administered. Patients were scanned after 60 min from ${}^{18}$F-FDG administration (uptake time), after hydration (500 mL water) and after voiding the bladder. Images were acquired on cross-calibrated GE Discovery MI, Discovery STE, Discovery 710 (General Electric, Milwaukee, WI, USA), acquisition time was 2 min per bed position. In summary, the acquisition protocol was performed according to the European Association of Nuclear Medicine (EANM) procedure guidelines [15].

### 2.4. Region Of Interest (ROI) Outlining

After that CT and PET studies were retrieved from PACS, manual segmentation of suspected local recurrence of rectal cancer was performed. As regards CT series, portal venous phase represents the best one to evaluate the presence of local recurrence due to the optimal impregnation of bowel walls. Therefore, it has been considered the one of choice for the segmentation of suspected local recurrence [16,17].

One expert radiologist (A.C.) with more than 10 years of experience in gastro-intestinal imaging contoured the target areas slice by slice, using ImageJ v1.53 (https://imagej.nih.gov/ij, accessed on 27 August 2022), a Java-based public-domain software [18].

As regards PET series, one nuclear medicine physician (P.C.) with more than 20 years of experience in PET reading outlined the ROIs using Aliza Medical Imaging 1.98.18 (https://www.aliza-dicom-viewer.com/, accessed on 2 September 2022) on the fused PET/CT images. Then, ROIs were reported at the original PET series resolution. Segmentation was performed according to recommendations of the Joint EANM and Society of Nuclear Medicine and Molecular Imaging (SNMMI) [19]. A fixed threshold (drawn on 40/50% of maximum Standardized Uptake Value, SUV) was drawn at first, followed by adjustments and corrections performed by the expert PET reader, due to the frequent presence of strongly heterogeneous lesions in the population in study.

### 2.5. Radiomic Feature Generation

The Radiomic Features (RFs) were generated, separately, from CT series during the venous contrast enhancement phase and SUV series. For RF computation, we adopted the method proposed in [20], and exploited in subsequent studies [11,21,22], LARC [21] included. The method is primarily based on the computation of 12 local first-order features (i.e., mean, median, kurtosis, skewness, entropy, uniformity, interquartile range, coefficient of variation, standard deviation, median absolute deviation, mean, and median of the last decile), calculated by assigning each ROI image pixel a feature computed on a surrounding window. To this aim, we adopted a square window, adaptively established to investigate a portion of 5 mm side length in CT and 10 mm side length in PET to have, for both the modalities’ resolution, the minimum significant number of pixels to be analyzed. The procedure provided us with 12 parametric maps of local first-order features. After that, the same previously mentioned 12 first-order features were computed on the global all-slice distributions of each first-order feature parametric map. Accordingly, in the end we had 144 RFs. All the procedures performed in this study were implemented in MatLab${}^{©}$ (R2021b v.9.11, The MathWorks, Natick, MA, USA).

### 2.6. Discrimination Study

Two discrimination studies were performed separately by CT and PET imaging modalities, looking for single RFs which individually allow separating the groups into LRRC (the positive class) and NO LRRC (the negative class); to this aim, we designed a two-stage procedure. In the first stage, after RFs standardization, the Least Absolute Shrinkage and Selection Operator (LASSO) method was utilized to select the most relevant RFs, by exploiting five-fold Cross-Validation (CV) at the minimum CV error rule, and weighing each sample by its prior probability. Then, a univariate analysis was carried out on the RFs coming through the first stage, by exploiting the Wilcoxon rank-sum test (*p* < 0.05) with Holm–Bonferroni correction to rank the RFs’ discrimination capability. The latter has been visually evaluated through boxplots and single group variances ($\sigma^{2}$) have been measured. In addition, Receiver Operating Characteristics (ROC) curves have been plotted and quantitatively evaluated through the Area Under the Curve (AUC). To assess the performance of the model, we also considered Sensitivity (SN), Specificity (SP), and Informedness (I) = SN + SP − 1. In the view of employing a combination of RFs in a future predictive study, each discriminant RF is herein reported and discussed.

## 3. Results

### 3.1. Clinical Characteristics of the Study Population

The clinical characteristics of the study population are summarized in Table 1. In addition, Table 2 reports the ROIs’ size, in cm${}^{3}$, for both CT and PET/CT images, within the two groups.

### 3.2. Discrimination of NO LRRC from LRRC Groups by CT Imaging

As regards the discrimination study by CT imaging, two RFs were selected by LASSO, the median of the last decile referred to the parametric maps of local kurtosis (K-M${}_{90th}$), and the median absolute deviation computed on the parametric maps of local skewness (S-MAD), and both of them resulted to be significant at univariate statistical analysis, with *p* = 0.007 and *p* = 0.022, respectively. In this regard, Figure 2a,b shows the boxplots of K-M${}_{90th}$ (a) and S-MAD (b), which both retain a higher value for LRRC group. In addition, K-M${}_{90th}$ shows higher variance in the LRRC group ($\sigma_{K-M_{90th}}^{2}$ = 1.09) than in the NO LRRC one ($\sigma_{K-M_{90th}}^{2}$ = 0.30), whilst S-MAD has a slightly higher variance in NO LRRC group ($\sigma_{S-MAD}^{2}$ = 0.91) than in LRRC one ($\sigma_{S-MAD}^{2}$ = 0.84). Figure 2c reports the ROC curve of K-M${}_{90th}$ and S-MAD, whilst related metrics are reported in Table 3. In particular, K-M${}_{90th}$ yields AUC = 0.82 (95% C.I. 0.50–0.97), SN = 57%, and SP = 100%, corresponding to I = 0.57, whilst S-MAD yields AUC = 0.76 (95% C.I. 0.53–0.97), SN = 67%, and SP = 71%, corresponding to I = 0.38.

**Figure 2 jcm-12-02058-f002:**
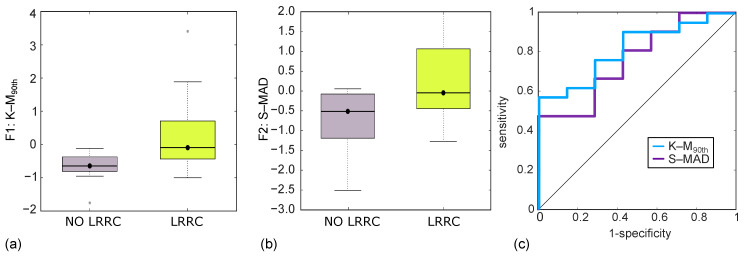
Boxplots of the discrimination achieved by CT imaging, between NO LRRC and LRRC groups, through K-M${}_{90th}$ (**a**) and S-MAD (**b**), and their ROC curve (**c**), yielding AUC = 0.82 (95% C.I. 0.50–0.97), SN = 57%, and SP = 100%, with I = 0.57 for K-M${}_{90th}$ and AUC = 0.76 (95% C.I. 0.53–0.97), SN = 67%, and SP = 71%, with I = 0.38 for S-MAD.

**Table 1 jcm-12-02058-t001:** Patients’ primary rectal cancer and pelvic recurrence characteristics.

	Total (n = 57)	Recurrence (n = 33)	No Recurrence (n = 24)
Age, years (median, IQR)	63 (52–71)	56 (50–68)	68 (64–72)
Male gender	34 (60%)	17 (52%)	17 (71%)
Tumor distance from AV, cm (median, IQR)	6 (4–9)	6 (4–10)	6 (4–9)
Neoadjuvant chemotherapy and/or radiotherapy	46 (81%)	23 (70%)	23 (96%)
Surgical procedure			
Sphincter preserving ${}^{a}$	42 (74%)	22 (67%)	20 (83%)
No sphincter preserving ${}^{b}$	15 (26%)	11 (33%)	4 (17%)
Histologic grade			
well/moderately differentiated	42 (74%)	22 (67%)	20 (83%)
poorly/undifferentiated	15 (26%)	11 (33%)	4 (17%)
Lymphovascular and/or perineural invasion	28 (49%)	22 (67%)	6 (24%)
Lateral pelvic nodes positive at MRI	4 (7%)	3 (9%)	1 (4%)
Pathologic stage			
pT1–2N0	2 (3%)	2 (7%)	0 (0%)
pT3–4N0	2 (3%)	1 (3%)	1 (4%)
pT1–4N+	5 (8%)	5 (15%)	0 (0%)
ypT0N0	4 (6%)	1 (3%)	3 (12.5%)
ypT1–2N0	13 (25%)	2 (6%)	11 (46%)
ypT3–4N0	11 (20%)	8 (24%)	3 (12.5%)
ypT1–4N+	17 (30%)	12 (36%)	5 (21%)
T any N any M1	3 (5%)	2 (6%)	1 (4%)
TRG sec. AJCC (n = 46)			
0	5 (11%)	1 (4%)	4 (17%)
1	5 (11%)	1 (4%)	4 (17%)
2	12 (26%)	6 (27%)	6 (27%)
3	24 (52%)	15 (65%)	9 (39%)
Anastomotic leakage (n = 42)	5 (12%)	3 (14%)	2 (10%)
Adjuvant chemotherapy	28 (49%)	21 (63%)	7 (29%)
Median time between primary resection and diagnosis of suspected LRRC, months (median, IQR)	13 (8–28)	13 (8–28)	14 (9–31)
Available imaging for radiomic analysis			
CT	28 (49%)	21 (64%)	7 (24%)
PET/CT	56 (98%)	32 (97%)	24 (100%)
Biopsy	39 (68%)	33 (100%)	6 (25%)
LRRC classification			
Central	-	11 (34%)	-
Sidewall	-	7 (21%)	-
Sacral	-	9 (27%)	-
Composite	-	6 (18%)	-
Multimodality salvage treatment LRRC			
Surgery	-	18 (55%)	-
CHT/RT	-	17 (52%)	-

IQR: interquartile range; AV: anal verge; MRI: magnetic resonance imaging; TRG: tumor regression grade; AJCC: American Joint Committee on Cancer; LRRC: local recurrence of rectal cancer; CT: computed tomography; PET: positron emission tomography; CHT: chemotherapy; RT: radiotherapy. ^*a*^ = included low anterior resection (LAR), proctectomy with coloanal anastomosis, total proctocolectomy with ileoanal pouch anastomosis (IPAA); ^*b*^ = included abdominal perineal resection (APR), Hartmann’s procedure.

**Table 2 jcm-12-02058-t002:** ROIs’ size, in cm${}^{3}$, for both CT and PET/CT images, within LRRC and NO LRRC groups.

	All Patients	LRRC Group	NO LRRC Group
**CT**			
Median, cm${}^{3}$	25.31	26.44	24.18
Range, cm${}^{3}$	[0.45–531.43]	[0.70–56.98]	[0.45–531.43]
**PET/CT**			
Median, cm${}^{3}$	9.32	9.14	9.32
Range, cm${}^{3}$	[0.98–189.97]	[0.98–189.97]	[1.91–54.73]

**Table 3 jcm-12-02058-t003:** Performance of the RFs finally selected in the CT discrimination study.

Rank	RF	AUC	SN	SP	I
1	K-M${}_{90th}$	0.82	57%	100%	0.57
2	S-MAD	0.76	67%	71%	0.38

RF: radiomic features; AUC = Area under the curve; SP: specificty; SN: sensitivity; I: informedness.

### 3.3. Discrimination of NO LRRC from LRRC Groups by PET/CT Imaging

As regards the discrimination study by PET imaging, five RFs were selected by LASSO and all of them resulted statistically significant at univariate analysis, with *p* < 0.017. In particular, the five RFs were the coefficient of variation computed on the parametric maps of local uniformity (U-CV, *p* = 0.005) and local median (M-CV, *p* = 0.006), the interquartile range referred to the parametric maps of local entropy (E-IQR, *p* = 0.006) and local median (M-IQR, *p* = 0.010), and S-MAD (*p* = 0.017), which is the same RF resulting from the study based on CT imaging.

Figure 3a–e shows the boxplots of the five RFs, where each of them shows higher values for LRRC group, as well as higher single group variances (ranging within [1.03–1.26] for LRRC group against [0.45–0.76] for NO LRRC group). In addition, Figure 3f also reports the ROC curves of the five RFs, whilst Table 4 resumes the main ROC-related metrics. Overall, AUC range was within [0.67–0.70], whilst SN = [56–63] %, SP= [71–79] %, and I = [0.30–0.39].

**Table 4 jcm-12-02058-t004:** Performance of the RFs finally selected in the PET/CT discrimination study.

Rank	RF	AUC	SN	SP	I
1	U-CV	0.70	75%	59%	0.34
2	M-CV	0.70	75%	63%	0.38
3	E-IQR	0.70	79%	56%	0.35
4	M-IQR	0.68	83%	56%	0.39
5	S-MAD	0.67	71%	59%	0.30

RF: radiomic features; AUC = area under the curve; SP: specificity; SN: sensitivity; I: informedness.

## 4. Discussion

In this study, we detected different RFs from post-operative CT and PET/CT imaging which enable significant and clear discriminations of the NO LRRC group from the LRRC one. It is worth noting that the detection of the two groups relies upon single RFs, this strengthening the statistical significance of results, being the ratio between the smallest group size (i.e., 7 NO LRRC for CT and 24 NO LRRC for PET/CT) and the RF (i.e., one) maximum [23].

The discriminatory study performed on PET/CT led to a number of RFs (i.e., five) higher than on CT, where two RFs only yielded significant discriminations between groups. Altogether, PET/CT imaging yielded stronger statistical significances than CT, as shown by almost all *p*-values being lower than that of the most significant CT-based RF (i.e., K-M${}_{90th}$, *p* = 0.007). Even S-MAD, the last-ranked PET/CT-based RF (*p* = 0.017), was more significant than S-MAD in CT (*p* = 0.022).

On the one hand, PET/CT imaging was expected to have a better accuracy than CT in detecting local recurrence, and to be capable to increase readers’ confidence levels reducing the number of equivocal cases [24,25]. Notably, all this was confirmed by the radiomic analysis. On the other hand, the high CT image resolution and the large volumes given by the ROIs submitted to radiomic analysis allowed identifying, even from the CT study, significant RFs able to detect those tissue properties claiming the presence of LRRC. Not least, our RFs, designed and conceived to perform a local analysis of small tissue patches, favoured the early detection of local tissue inhomogeneities, even within small volumes, thus incrementing the global sensitivity of the radiomic analysis.

As regards the meaning of these RFs, some of them coming from PET/CT study, represent different measures of the degree of local tissue heterogeneity and its variability within the ROI volume (e.g., E-IQR, M-IQR, U-CV, M-CV). Other RFs arising from PET/CT or CT studies quantify the departure of image values within the ROI volume, from a Gaussian process (e.g., K-M${}_{90th}$, S-MAD), this early depicting, according to a previous report [21], a dynamic behavior of evolving tissues, consistent with the presence of LRRC.

One RF, S-MAD, had significant results in both PET/CT and CT studies, somehow recording one tissue behavior represented by different physical measures exploiting different tissue properties. However, the implications of this finding have yet to be explored. Basically, S-MAD quantifies the dispersion around the median of the local skewness measurements, where higher S-MAD values correspond to areas showing a wide inhomogeneity of underlying image values, characterized by mostly higher local values, whether they refer to SUV (PET/CT) or Hounsfield unit (H.U.) (CT). Similarly, lower S-MAD values describe areas with a wide inhomogeneity, with mostly lower values. This tendency might be explained by the fact that high S-MAD values may represent local recurrence, being characterized by contrast-enhanced heterogeneous tissue, whilst low S-MAD ones, especially in patients belonging to the NO LRRC group, may characterize what was suspected to be pathological tissue, eventually proved to be a non-neoplastic consequence of surgery (such as regenerative or fibrotic tissue) which, although heterogeneous on CT scan, most often appears as scarcely enhancing.

To date, radiomic models have been tested in patients with LRRC in only one study [12]. Chen and colleagues used a MRI-based radiomic model to predict local recurrence at the site of anastomosis, in a retrospective case series of 80 patients with clinically suspected tumor relapse, 11 of which had confirmed local recurrence at the site of anastomosis [12]. The study reported good predictive performance in the validation set, especially for the so-called combined model, arising from three different MRI sequences. However, this study presents major flaws from both scientifical and clinical points of view. First, the study trained the model on an oversampled dataset starting from 40 samples (constituting the training subset), where only 5 reported local recurrence. Unfortunately, such a small distribution cannot be reliably oversampled, this raising several doubts thwarting the statistical significance of results. In addition, single-sequence predictive models, and even more, the combined one, relied on a number of radiomic features that was incredibly high if compared to the sample size of the dataset, this weakening the predictive role of the results. Moreover, from a clinical point of view, surveillance proctosigmoidoscopy with or without endorectal ultrasound, in addition to interval colonoscopy, is typically recommended after proctectomy in patients with rectal cancer for detecting local recurrence [8]. In addition to this, in case of suspected local recurrence, a histological examination of biopsy samples during proctoscopy may be sufficient for ruling out a recurrence, without complications or discomfort for patient, thus weakening the need for radiomic analysis. Instead, the identification of pelvic recurrence is much more complex because LRRC is often located within pre-existing fibrosis, due to the primary operation, infectious complications, and prior radiotherapy.

As far as the diagnostic imaging is concerned, the reported accuracy of CT in detecting LRRC ranges from 68% to 76% [26]. Then, compared with CT, MRI can more accurately detect LRRC due to its excellent soft-tissue resolution. Indeed, although an area that increases in size, which has an asymmetric, heterogeneous, and marked contrast enhancement appearance, with an invasive behaviour, is suspicious for LRRC; small growing tumors within fibrotic scar tissues remain difficult to detect [27,28]. Actually, even PET/TC has some limitations in detecting small lesions and evaluating mucinous tumor recurrence because mucinous adenocarcinomas have poor FDG uptake. In addition, false positives may occur in areas of post-operative infectious or inflammatory scar tissue [28,29]. For these reasons, to date, only repeated imaging and CT-guided biopsy sampling can confirm the presence of growing mass.

Our work is the first that has shown how CT and PET/CT radiomics allow accurately differentiating scar tissue from both anastomotic and extraluminal relapses. However, some limitations have to be considered. First, it was a retrospective single-institutional study. Second, the limited sample size of the study population did not allow (i) including clinical variables within the analysis and, (ii) developing a predictive model, but only performing a discriminatory study. Therefore, wider patient cohorts are needed to confirm the predictive value of the RFs selected from CT and PET/CT. However, the very good discriminative capability achieved using single RFs from both imaging modalities, represent a solid premise encouraging the future development of predictive radiomic signatures from both single-modality and hybrid (i.e., CT combined with PET/CT) imaging.

## 5. Conclusions

In conclusion, the radiomic analysis of contrast-enhanced-CT and PET/CT images could be an additional useful tool for discriminating LRRC from scar tissue in patients with rectal cancer after complete surgical tumor removal. CT and PET/CT radiomics may favor a tailored approach that allows selecting patients for biopsy more accurately.

## Figures and Tables

**Figure 1 jcm-12-02058-f001:**
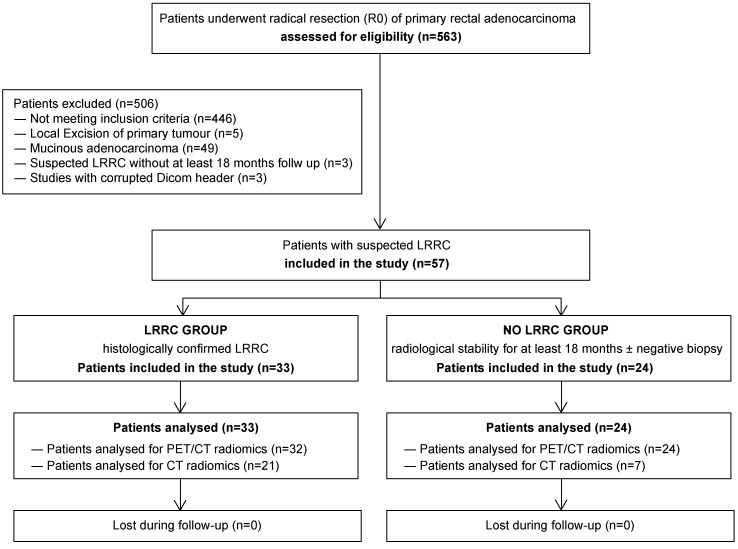
Flowchart of patients’ inclusion.

**Figure 3 jcm-12-02058-f003:**
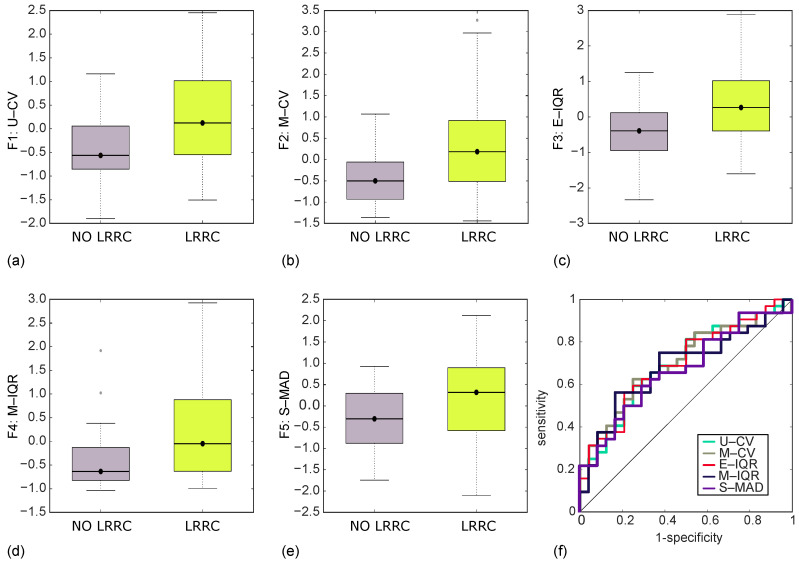
Boxplots of the discrimination achieved by PET/CT imaging, between NO LRRC and LRRC groups, through U-CV (**a**), M-CV (**b**), E-IQR (**c**), M-IQR (**d**), and S-MAD (**e**), and their ROC curve (**f**), yielding AUC = [0.67–0.70], SN = [56–63]%, SP = [71–79]%, and I = [0.30–0.39].

## Data Availability

The data are not available because of patient privacy.

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
