# Peer review of "Radiomic Features from Post-Operative 18F-FDG PET/CT and CT Imaging Associated with Locally Recurrent Rectal Cancer: Preliminary Findings"

_jcm, 2023, doi:10.3390/jcm12052058_

Round 1
Reviewer 1 Report
The paper deals a current topic and the aim is extremely interesting. In the light of new radiomics trend, data that predict LRRC diagnosis could be extremely useful.
The work present a satisfactory structure but there are some limitations, already highlighted by the authors in discussion, that could affect his own newly and quality.
The main limitations are the retrospective structure of the study and the limited number of included patients. In particular, this last point ma make the interesting results obtained by the authors less significant.
Probably the next step should be to extend the pool of patients although obviously not easy to implement and validate the real benefits of this approach.
In any case, the discussion and analysis of the data proposed by the authors are interesting and stimulating for future developments in clinical practice.
Satisfactory images and tables.
Adequate reference section
Author Response
RESPONSES TO THE COMMENTS
Hereafter, “C#” stands for Reviewer’s “Comment” and “R#” are our Replies (in red). Similarly, the changes in the manuscript are reported in red as well.
REVIEWER #1
General comments
The paper deals a current topic and the aim is extremely interesting. In the light of new radiomics trend, data that predict LRRC diagnosis could be extremely useful.
The work present a satisfactory structure but there are some limitations, already highlighted by the authors in discussion, that could affect his own newly and quality.
Specific comments
[C1.1] The main limitations are the retrospective structure of the study and the limited number of included patients. In particular, this last point ma make the interesting results obtained by the authors less significant. Probably the next step should be to extend the pool of patients although obviously not easy to implement and validate the real benefits of this approach. In any case, the discussion and analysis of the data proposed by the authors are interesting and stimulating for future developments in clinical practice.
[R1.1] We thank the reviewer for his/her warm appreciation. Actually, we thought at this study as a preliminary one, for investigating the potential contribution of the radiomic analysis in the cohort of patients with local recurrence of rectal cancer. As the reviewer guessed, based on our encouraging findings, we are planning a dedicated prospective and larger patient enrolment, which can allow developing and validating a predictive model.
[C1.2] Satisfactory images and tables. Adequate reference section.
[R1.2] Many thanks.
Reviewer 2 Report
Overall, the study appears to be well-conducted and provides valuable insights into the potential use of radiomics for predicting pelvic recurrence in patients with rectal cancer. The authors have used appropriate methods to extract radiomic features from CT and PET/CT images and have employed appropriate statistical analyses to evaluate the predictive performance of these features. The study is also well-written and structured.
However, there are a few major limitations in this study:
- Limited sample size: The study was conducted on a relatively small sample size of 57 patients, which may limit the generalizability of the results.
- Single-center study: The study was conducted in a single center, which may limit the generalizability of the findings to other centers with different patient populations and imaging protocols.
- Retrospective study design: The study design was retrospective, which may introduce bias and limit the ability to control for confounding variables.
- Lack of external validation: The study did not include external validation of the radiomics models on an independent dataset, which may limit the reliability of the findings.
- Lack of information on inter-reader variability: The study did not report information on inter-reader variability, which may affect the reproducibility and generalizability of the radiomics models.
- Lack of consideration of potential confounding factors: The study did not consider potential confounding factors, such as the timing of imaging, the type of contrast agent used, and the presence of other comorbidities, which may affect the performance of the radiomics models.
Author Response
RESPONSES TO THE COMMENTS
Hereafter, “C#” stands for Reviewer’s “Comment” and “R#” are our Replies (in red). Similarly, the changes in the manuscript are reported in red as well.REVIEWER #2
General comments
Overall, the study appears to be well-conducted and provides valuable insights into the potential use of radiomics for predicting pelvic recurrence in patients with rectal cancer. The authors have used appropriate methods to extract radiomic features from CT and PET/CT images and have employed appropriate statistical analyses to evaluate the predictive performance of these features. The study is also well-written and structured.
Specific comments
However, there are a few major limitations in this study:
[C2.1] Limited sample size: The study was conducted on a relatively small sample size of 57 patients, which may limit the generalizability of the results.
[R2.1] We are aware of this limitation, as it was mentioned in the dedicated paragraph of Discussions. For this reason, we could just perform a discrimination study. However, although on a limited sample size, the discrimination between patients with and without local recurrence of rectal cancer (LRRC) is achieved with single radiomic features, thus minimizing the risk of overfitting and favouring a perspective generalizability of results, also due to the very good statistical significance achieved.
[C2.2] Single-center study: The study was conducted in a single center, which may limit the generalizability of the findings to other centers with different patient populations and imaging protocols.
[R2.2] See also [R2.1]. We agree with the reviewer that a multi-centre study will strengthen our findings, and we are planning a multicentre study, also to widen the study population. As a matter of fact, our pipeline considers differences in imaging protocols (see “Radiomic feature generation” section), being based on adaptive methods and, as such, this makes it ready for multi-centre analysis.
[C2.3] Retrospective study design: The study design was retrospective, which may introduce bias and limit the ability to control for confounding variables.
[R2.3] We did not include clinical parameters, not only because of the limited sample size, but also to avoid the confounding effects of variables acquired, retrospectively, in non-standardized ways. According to reviewer concern, this has been clarified in the manuscript by rephrasing the following sentence:
Lines(L)322-325: Second, the limited sample size of the study population did not allow (i) including clinical variables within the analysis and, (ii) developing a predictive model, but only performing a discriminatory study. Therefore, […]
In addition, all imaging depending factors have been considered while designing the pipeline, by developing adaptive methods for image processing. These methods would act equally also in case of patient image series coming from a prospective enrolment.
[C2.4] Lack of external validation: The study did not include external validation of the radiomics models on an independent dataset, which may limit the reliability of the findings.
[R2.4] See [R2.1]. The external validation is out of purposes of a discrimination study, which can be considered as the early as well as necessary stage of a predictive study, where the external validation is mandatory. Nonetheless, the encouraging results we achieved will be exploited in the prospective study being planned, to enlarge the patient cohort, that will be used to develop, and validate a predictive model. A further specification has been added in the manuscript by rephrasing the sentence reported in [R2.3].
[C2.5] Lack of information on inter-reader variability: The study did not report information on inter-reader variability, which may affect the reproducibility and generalizability of the radiomics models.
[R2.5] Differently from what happens with other radiomic features, our features are local, meaning that pixel of the regions of interest, including those nearby borders, are not used based on their single value, but on their neighbourhoods’. This makes our feature robust, by design, to possible differences in segmentation.
[C2.6] Lack of consideration of potential confounding factors: The study did not consider potential confounding factors, such as the timing of imaging, the type of contrast agent used, and the presence of other comorbidities, which may affect the performance of the radiomics models.
[R2.6] We do not expect that other confounding factors but those treated during the image processing pipeline, may substantially impact on the findings of the radiomic study. As regards the timing of imaging CT study is the first to be performed in case of suspected LRRC and PET/CT soon after. In addition, being the radiomic studies performed on CT and PET/CT independent, any timing difference between the two examinations cannot affect the results of single studies. As for the contrast agent, all patients included underwent the same injection protocol of iodic contrast agent, thus excluding it could be a source of variation of image properties among our cohort. Finally, to our best knowledge, there is no reason to hypothesize that other comorbidities, such as diabetes, cardiopathy, etc., might affect CT and PET/CT image properties at the site of suspected or confirmed LRRC, and the radiomic models accordingly.

Round 2
Reviewer 2 Report
responses are acceptable.
Author Response
Thank you for your appreciation